# Effectiveness, acceptability, and completeness and quality of intervention reporting of psychological interventions for people with dementia or mild cognitive impairment: protocol for a mixed-methods systematic review

Frida Svedin ,[1] Oscar Blomberg,[1] Paul Farrand,[2] Anders Brantnell,[1,3] Louise von Essen,[1] Anna Cristina Åberg,[4,5] Joanne Woodford [1]

For numbered affiliations see end of article.

**Correspondence to**
Dr Joanne Woodford;
joanne.woodford@kbh.uu.se

## ABSTRACT

**Introduction** Mental health difficulties such as anxiety and depression have negative impacts on psychological well-being and are common in people with dementia and mild cognitive impairment. However, access to psychological treatments is limited. This mixed-method systematic review will: (1) examine the effectiveness of psychological interventions to improve mental health and psychological well-being in people with dementia or mild cognitive impairment; (2) examine the effectiveness of these psychological interventions to improve mental health and psychological well-being in informal caregivers; (3) examine potential clinical and methodological moderators associated with effectiveness; (4) explore factors associated with the acceptability of psychological interventions from the perspective of key stakeholders; and (5) examine the completeness and quality of intervention reporting.

**Methods and analysis** Electronic databases (ASSIA, CENTRAL, CINAHL, EMBASE, PsycINFO and MEDLINE) will be systematically searched and supplemented with expert contact, reference and citation checking, and grey literature searches. If possible, we will conduct a meta-analysis to examine the overall effectiveness of psychological interventions to improve mental health and psychological well-being in people with dementia or mild cognitive impairment and their informal caregivers; and examine potential clinical and methodological moderators associated with effectiveness. We will conduct a deductive framework synthesis, informed by the theoretical framework of acceptability, to explore factors associated with the acceptability of psychological interventions from the perspective of key stakeholders. In accordance with Joanna Briggs Institute guidance, we will adopt a convergent segregated approach to data synthesis and integration of quantitative and qualitative findings. We will examine the completeness and quality of intervention reporting according to the Template for Intervention Description and Replication checklist and guide.

**Ethics and dissemination** No primary data will be collected, and therefore, ethical approval is not required.

## STRENGTHS AND LIMITATIONS OF THIS STUDY

⇒ This review will extend existing literature by including novel aspects such as narrowing the classification and definition of eligible psychological interventions, excluding studies with a high risk of bias, and exploring intervention acceptability, and the completeness of intervention reporting.

⇒ We will adopt a mixed-method approach informed by the convergent segregated approach, allowing us to synthesise evidence from multiple sources to address review questions.

⇒ This review will follow quality standards informed by the New Cochrane Handbook for Systematic Reviews of Interventions, Preferred Reporting Items for Systematic Reviews and Meta-Analyses 2020 guidelines, and Enhancing Transparency in Reporting the Synthesis of Qualitative Research.

⇒ Despite the Critical Appraisal Skills Programme checklist being a recommended tool for assessing the quality of qualitative studies, it does not include a quality score rating system. This may lead to including low-quality qualitative studies.

⇒ Due to limited resources, only studies available in English and Swedish will be included.

Results will be disseminated through a peer-reviewed publication, academic conferences, and plain language summaries.

**PROSPERO registration number** CRD42023400514.

.

## INTRODUCTION

Dementia is a global public health priority,[1] with the number of people living with dementia estimated to rise from 50 million in 2016 to 152 million in 2050.[2] Dementia has

several significant adverse impacts, including increased mortality,[3] cognitive[4] and functional decline,[5] and poor quality of life.[6] Significant demands are placed on informal caregivers (ie, family and friends who provide unpaid care) and wider society.[7] People with dementia and mild cognitive impairment (MCI) commonly experience comorbid mental health difficulties, such as anxiety and depression[8–11] which are associated with exacerbation of cognitive decline, poor quality of life, sleep difficulties, and reduction in activities of daily living.[12] For people with dementia, the overall pooled prevalence is estimated to be 39% for both anxiety and for depression,[10] and for people with MCI 21% for depression[8] and 32% for anxiety.[9] Despite research suggesting psychological interventions are effective,[13–15] access to evidence-based psychological support for people with dementia is limited.[16 17]

Recent reviews suggest psychological interventions for anxiety and depression in dementia or MCI are probably effective.[13–15 18] However, these reviews have included a broad range of psychosocial interventions (eg, psychosocial interventions focusing on psychological and social factors, relaxation training therapies, music therapies, exercise, and massage) yielding large heterogeneity in intervention type. To facilitate subgroup analyses to explore the potential moderating effect of psychological intervention type, we intend to build on previous reviews by using a narrower classification of psychological interventions.[19] Using a narrower classification and potentially reducing clinical heterogeneity may facilitate identifying clinical moderators rarely investigated such as intervention delivery model (eg, individual, dyadic, group, or self-administered) and method of support (eg, face to face, internet, telephone, or video conference).

Further, previous reviews[13–15 18] have not examined psychological interventions for psychological well-being in people with dementia or MCI.[20] Such interventions may be of particular importance given the challenges experienced in diagnosing people with dementia and depression due to overlapping symptomology between late-life depression and dementia.[21] Difficulties are also experienced in diagnosing anxiety, given people with dementia or MCI may present with anxiety symptoms inconsistent with standardised diagnostic criteria.[22] Additionally, informal caregivers are at an increased risk of experiencing difficulties with their mental health and psychological well-being, for example, burden and depression,[23] and poor quality of life.[24] Given the dyadic interdependence between the care recipient and caregiver's psychological well-being,[25] it may be important to examine whether psychological interventions to improve mental health and psychological well-being in people with dementia or MCI also improve informal caregiver mental health and psychological well-being. In a previous review examining the effect of psychological treatments for anxiety and depression for dementia and MCI, a small number of studies were found that included informal caregiver outcomes.[14] However, how interventions aimed at improving psychological well-being or mental health-related quality of life in people with dementia and MCI may also impact informal caregiver mental health and psychological well-being warrants further investigation. Finally, existing reviews have included studies with non-randomised designs[13] or with a high risk of bias concerning randomisation procedures.[15] Including these studies may result in an overestimation of treatment effect.[26 27] Consequently, there is a need to conduct a systematic review excluding studies of low methodological quality.

In addition to the aforementioned gaps in the evidence base, people with dementia or MCI commonly experience barriers to accessing psychological support. Barriers include factors related to the individual (eg, lack of time for treatment or geographical distance), the provider (eg, lack of willingness to diagnose and treat mental health difficulties) and the healthcare system (eg, limited availability of specialty mental health providers).[28] Barriers commonly experienced by people with dementia or MCI include social stigmatisation at the individual level,[29] and lack of knowledge of mental health problems among healthcare providers[17 30] at the provider level. An additional access barrier is intervention acceptability, that is, the anticipated or experienced cognitive and emotional response to an intervention from the perspective of those receiving and delivering the intervention.[31] For example, lack of motivation and low patient interest represent common barriers to intervention access among older adults.[32] It is essential to consider acceptability when developing, evaluating, and implementing healthcare interventions[31 33] given associations with intervention adherence[31] and positive clinical outcomes.[34] However, to date, little is known about the acceptability of psychological interventions for people with dementia or MCI, and other important stakeholders (eg, informal caregivers and healthcare professionals). Considering factors associated with acceptability[35] may help improve treatment access and enhance the implementation potential of future developed interventions by increasing intervention acceptance and relevance from the perspective of those receiving and delivering/implementing the intervention.

An additional barrier to treatment access may relate to the completeness and quality of the reporting of psychological interventions in the research literature. The completeness and quality of published intervention descriptions (ie, descriptions of evaluated interventions that are detailed enough to facilitate replication) are commonly poor.[36 37] Consequently, reliable intervention replication and/or implementation is difficult.[36] Ensuring the transparency and completeness of intervention reporting may facilitate the adoption of interventions by clinicians, informal caregivers, patients, and policy decision-makers,[38] and is essential for implementing interventions in practice.[39] Existing reviews suggest intervention components of psychological interventions for people with dementia or MCI are poorly reported.[14] However, to the best of our knowledge, there is no comprehensive review describing the quality and

completeness of intervention reporting of psychological interventions for people with dementia or MCI.

In summary, this mixed-methods systematic review will address the aforementioned gaps in the literature by: (1) using a narrower and commonly used classification of psychological interventions,[19] to reduce clinical heterogeneity and facilitate comparison of findings and exploration of potential clinical moderators; (2) examining the effectiveness of psychological interventions for people with dementia or MCI on general psychological well-being and mental health-related quality of life; (3) examining the effectiveness of these psychological interventions to improve the mental health and psychological well-being in informal caregivers; (4) excluding studies with non-randomised and uncontrolled designs and randomised controlled trials (RCTs) with randomisation procedures providing a high risk of bias, in line with the Cochrane Collaboration's Risk of Bias tool 2.0 (RoB 2.0)[40]; (5) exploring intervention acceptability from the perspective of important stakeholders (eg, people with dementia or MCI, informal caregivers and healthcare professionals); and (6) examining intervention reporting quality and completeness.

## Objectives

This mixed-method systematic review will: (1) examine the effectiveness of psychological interventions to improve mental health and psychological well-being in people with dementia or MCI; (2) examine the effectiveness of these psychological interventions to improve mental health and psychological well-being in informal caregivers; (3) examine potential clinical and methodological moderators associated with effectiveness; (4) explore factors associated with the acceptability of psychological interventions from the perspective of key stakeholders; and (5) examine the completeness and quality of intervention reporting.

## METHODS AND ANALYSIS

The review will adopt a mixed-methods approach with the protocol reported in accordance with the Preferred Reporting Items for Systematic Reviews and Meta-Analyses Protocols (PRISMA) statement[40] (online supplemental appendix 1) and Enhancing Transparency in Reporting the Synthesis of Qualitative Research (ENTREQ) statement (online supplemental appendix 1).[41] Study results will be reported following the PRISMA 2020 guidelines.[42] This protocol has been registered with the International Prospective Register of Systematic Reviews registration number: CRD42023400514 and is partially informed by a previous review protocol.[43] Centre of Reviews and Dissemination guidance for conducting systematic reviews,[44] the New Cochrane Handbook for Systematic Reviews of Interventions,[45] and Joanna Briggs Institute (JBI) guidance for mixed-methods systematic reviews will be followed.[46]

## Inclusion and exclusion criteria
### Participants
Adults diagnosed with dementia or MCI in accordance with a validated diagnostic criterion (eg, the Diagnostic and Statistical Manual, International Classification of Diseases, other validated criteria) or recorded in medical records. Eligible dementia diagnoses include Alzheimer's disease and related dementias (eg, cerebrovascular/vascular disease, frontotemporal dementia, Lewy body disease or mixed). Adults diagnosed with dementia primarily due to HIV, Huntington's disease, Parkinson's disease, Prion disease, substance/medication use, traumatic brain injury, multiple sclerosis, or other medical conditions will be excluded. No restrictions will be placed on dementia severity or time since diagnosis. Community and institutional-dwelling persons with dementia or MCI will be eligible for inclusion. Persons with comorbid severe mental health problems (eg, bipolar affective disorder, post-traumatic stress disorder, or psychosis) will be excluded.

### Interventions
Psychological intervention eligibility will be based on a classification of evidence-based psychological interventions commonly used in systematic reviews of psychological interventions.[19] These include cognitive behavioural therapy, behavioural activation therapy, problem-solving therapy, third-wave cognitive behavioural therapies, interpersonal psychotherapy, psychodynamic therapy, nondirective supportive therapy, and life review therapy. Such interventions include specific therapeutic techniques designed to improve mental health and psychological well-being (eg, anxiety, depression, mental health-related quality of life, and psychological distress). Different intervention delivery modes (eg, individual, group, a person with dementia-informal caregiver dyad, and self-help), methods of support (eg, email, face-to-face, internet, telephone, and video conference), intervention setting and professional background of the person supporting or delivering the intervention (eg, nurses, psychologists, and psychological practitioners) will be eligible for inclusion.

### Comparators
Eligible study designs will allow for treatment effects to be isolated[47] with eligible comparator conditions based on standard definitions[48] and include: (1) no-treatment control; (2) wait-list control; (3) treatment as usual (TAU); (4) non-specific factors component control; (5) specific factors component control; and (6) active comparator.

### Outcomes
#### Quantitative
Eligible studies will include one or more self-report, clinician, or proxy reports by an informal caregiver standardised primary outcome measurement of mental health or psychological well-being in the person with dementia or MCI, including: (1) anxiety; (2) depression; (3) mental health-related quality of life; (4); and psychological distress (defined as a measurement of non-symptoms associated with anxiety, depression, and stress);[49] and (5) psychological well-being.

### Qualitative

Qualitative studies eligible for inclusion will be: (1) studies that precede included RCTs conducted to inform intervention development and (2) embedded studies within included RCTs that explore the acceptability (eg, attitudes, barriers and facilitators to intervention use, perceived burden, perceived benefits, perceived effectiveness, perceived relevance, satisfaction, and understanding) of included psychological interventions from the perspective of important stakeholders (eg, people with dementia or MCI, informal caregivers, and healthcare professionals). If mixed-methods studies are identified using quantitative measures of intervention acceptability (eg, the Treatment Acceptability/Adherence Scale),[50] data will be extracted and triangulated with the qualitative findings.

### Study designs
### Quantitative

To examine intervention effectiveness, RCTs and cluster RCTs will be eligible for inclusion. Non-RCTs will be excluded. Given the likelihood that including studies of low quality,[51–54] or without a proper RCT design inflates the effect size,[26 27] studies rated as high risk of bias regarding allocation and concealment randomisation processes in accordance with the RoB 2 tool[55] will be excluded.

### Qualitative

To explore acceptability, studies using any type of qualitative research study design exploring the acceptability of interventions from the perspective of key stakeholders (eg, people with dementia or MCI, informal caregivers, and healthcare professionals) will be eligible for inclusion.

### Search methods
### Electronic searches

A search strategy based on medical subject headings (MeSH) will be carried out in the following relevant international electronic databases: ASSIA, CENTRAL, CINAHL, EMBASE, PsycINFO, and MEDLINE. A draft of the MEDLINE search strategy can be found in online supplemental appendix 2. Searches in the remaining electronic databases will be adapted to the syntax and subject headings. The search strategy was developed alongside a librarian at Uppsala University Library. Electronic databases will be searched from the date of inception up to the date the search was conducted.

### Other resources

Hand searches will supplement electronic searches (eg, expert contact, back-forward citation, and reference checking) to identify additional RCTs potentially eligible for inclusion. Hand searches will also be used to search for qualitative research preceding and embedded within included RCTs that explore the acceptability of included interventions. Grey literature will be searched using OpenGrey (http://www.opengrey.eu/).

No date restrictions will be imposed when searching electronic or other resources, however, due to resource limitations only papers published in English and Swedish will be included. Both published and unpublished studies will be eligible for inclusion, however, due to time constraints, we will not include full dissertations.

### Screening and data extraction
### Screening

Title and abstract screening, and full paper checking will be performed by two reviewers independently. Duplicate studies across searches will be identified and removed before title and abstract searching using EndNote V.X9 (Clarivate).[56] Titles and abstracts from retrieved citations will be screened independently by two reviewers using Rayyan.[57] Full papers of potentially eligible studies will be checked independently by two reviewers against the inclusion/exclusion criteria. The two reviewers will discuss discrepancies, with a third reviewer consulted if needed. Corresponding authors will be emailed to resolve questions about eligibility when necessary, with a follow-up email sent to corresponding authors who do not respond to the first email within 2 weeks. Reasons for exclusion will be recorded and reported in a PRISMA flow chart. A detailed exclusion table will be presented for each specific inclusion/exclusion criterion in accordance with the Population, Intervention, Comparison, Outcome, and Study design (PICOS) statement.

### Quantitative data extraction

Two reviewers will independently extract data from the included studies into Microsoft Excel. Reviewers will resolve disagreements by discussion, with a third reviewer adjudicating any disagreements. In case of any continued uncertainties, corresponding authors will be contacted by email, with a follow-up email for authors who do not respond to the first email within 2 weeks. The following data will be extracted:

Study characteristics: aims and objectives, country of origin, funding, identification features (ie, authors, citation details), inclusion/exclusion criteria, language, primary outcome measurements, publication type, and study design.

Participant characteristics: age, dementia subtype (eg, Alzheimer's disease cerebrovascular/vascular disease, frontotemporal dementia, Lewy body disease or mixed), how dementia diagnosis was established, educational status, ethnicity, gender, provision of support from an informal caregiver, severity of cognitive impairment and dementia, and time since dementia diagnosis.

Intervention characteristics: extracted as per the Template for Intervention Description and Replication (TIDieR) checklist[36] (table 1).

Recruitment characteristics: recruitment of individual persons with dementia or MCI or dyads (ie, the person with dementia or MCI and informal caregiver), recruitment setting, recruitment method, sampling method, and type of consent.

Participant flow: total number of people with dementia or MCI (and informal caregivers where applicable) invited,

**Table 1** The Template for Intervention Description and Replication (TIDieR) checklist and guide*

| Item number | TIDieR item | Brief description |
|---|---|---|
| 1 | Brief name | Brief name of intervention: the name or phrase that describes the intervention |
| 2 | Why | Rationale, theory or goal of the intervention |
| 3 | What | Physical or informational materials: provided to participants or used in intervention delivery or in training intervention providers |
| 4 | | Procedures, activities and processes: including enabling or support activities |
| 5 | Who provided | Intervention provider: including expertise, background and specific training given |
| 6 | How | Mode of delivery, for example, face-to-face, internet, telephone and whether provided individually or in a group |
| 7 | Where | Location: including any necessary infrastructure or relevant features |
| 8 | When and how much | Timing, duration and Intensity: number of times the intervention was delivered and over what period of time including the number of sessions, their schedule and their duration, intensity or dose |
| 9 | Tailoring | Tailoring of Intervention: if planned to be personalised, titrated or adapted, then describe what, why, when and how |
| 10 | Modification | Modification of intervention: if modified during the course of the study, describe the changes (what, why, when and how) |
| 11 | How well | Planned: if intervention adherence or fidelity was assessed, describe how and by whom, and if any strategies were used to maintain or improve fidelity |
| 12 | | Actual: if intervention adherence or fidelity was assessed, describe the extent to which the intervention was delivered as planned |

*Adapted from Hoffmann et al.[36]

screened, eligible and randomised, number randomised to each intervention arm. For each intervention arm and time point: number analysed and lost to follow-up, with number attrition ≤6 months and percentage attrition ≤6 months calculated.

Results: primary outcomes measurement (see the Outcomes section for primary outcomes).

For each primary outcome and intervention arm: time point, mean, SD (or SE or CI), and number of participants analysed.

Intervention acceptability: intervention and study dropout rates, adherence to the intervention, quantitative measures of intervention acceptability and satisfaction (eg, the Treatment Acceptability/Adherence Scale).[50]

### Qualitative data extraction

Qualitative data from included studies will be imported into NVivo[58] to facilitate analysis. The following data will be extracted:

Study characteristics: standard study information including identification features, publication type, funding, language, aims and objectives, country of origin, funding, identification features (ie, authors, citation details), language, and qualitative methodology.

Stakeholder characteristics: data describing the characteristics of important stakeholders (eg, people with dementia or MCI, informal caregivers, and healthcare professionals).

Acceptability: qualitative data exploring the acceptability of interventions from the perspective of important stakeholders (eg, attitudes, barriers and facilitators to intervention use, perceived burden, perceived benefits, perceived effectiveness, perceived relevance, satisfaction, and understanding).

### Quality appraisal
#### Quantitative
##### Risk of bias

Risk of bias for each included study will be assessed independently by two reviewers using the Cochrane Collaboration's Risk of Bias tool 2.0 (RoB 2.0).[55] Published material (eg, published paper, study protocol, and trial registration) and additional unpublished information from study authors will inform the assessment. We will contact study authors if there is missing relevant information or discrepancies between preregistered information (study protocol, trial registration) and the published results paper. The following domains will be assessed: (1) randomisation process, (2) deviations from intended interventions, (3) missing outcome data, (4) measurement of the outcome, and (5) selection of the reported result. Two reviewers will discuss rating discrepancies, if a consensus cannot be reached a third reviewer will be consulted. Each domain can be rated as of low risk of bias, some concerns, or of high risk of bias in accordance with RoB 2.0 guidance and results will be reported descriptively with the Robvis tool[59] used to visualise risk of bias.

### Confidence in evidence

The Grading of Recommendations Assessment, Development and Evaluation tool (GRADE)[60] will be used to assess the confidence in evidence for the primary outcomes (mental health and psychological well-being (eg, anxiety, depression, mental health-related quality of life, and psychological distress)) across studies.[61 62] Confidence in evidence will be rated as very low, low, moderate or high. RCTs receive an initial high rating but confidence in evidence is downgraded if rated as not having sufficient quality in any of the five domains: (1) within-study risk of bias; (2) directness of evidence; (3) heterogeneity; (4) precision of effect estimates; and (5) risk of publication bias.[60]

### Qualitative

Two researchers will independently assess the methodological quality of included qualitative studies using the Critical Appraisal Skills Programme (CASP) checklist for qualitative studies.[63] Rating discrepancies will be discussed with a third reviewer consulted if consensus is not reached. Results will be reported descriptively.

### Data synthesis

Informed by the convergent segregated approach, quantitative and qualitative analyses and syntheses will first be conducted separately, followed by an integration of evidence derived from both syntheses.[46] Quantitative data will be extracted into Microsoft Excel, narratively synthesised, and if data allow analysed using Comprehensive Meta-analysis (V.4). Qualitative data will be managed and analysed using NVivo.[58]

### Meta-analysis

If data permit, a meta-analysis will be conducted. Post-treatment between-group standardised mean effect sizes will be calculated (Hedges' g)[64] adopting a random effects model for:

▶ Outcomes relating to mental health and psychological well-being (eg, anxiety, depression, mental health-related quality of life, and psychological distress) separately for people with dementia or MCI.

▶ Outcomes relating to mental health and psychological well-being (eg, anxiety, depression, mental health-related quality of life, and psychological distress) separately for informal caregivers.

In studies with multiple time points, a primary endpoint of ≤6 months will be adopted, reducing the potential bias associated with short-term post-treatment effects and the risk of elevated effect sizes.[43 51 65 66] In accordance with the Cochrane Handbook,[45] studies where two intervention groups are eligible for inclusion, the control group sample size will be halved, and comparisons with each intervention group will be conducted separately. Where two control groups are compared with one intervention group, the intervention sample size will be halved and comparisons with each control group will be conducted separately.[67] A random-effects model will be used given anticipated heterogeneity due to variations in intervention and methodological components, and participant characteristics.[68] Cochran's Q statistic will be used to examine heterogeneity.[69] The proportion of total variability due to between-study heterogeneity will be measured using $I^2$ and the prediction interval will be adopted as an index of population dispersion.[70 71]

### Moderator analysis

If sufficient data from included RCTs allow, subgroup analyses will examine the potential moderating effects of:

▶ Intervention delivery mode (eg, individual, person with dementia-informal caregiver dyad, group, self-help).

▶ Length of follow-up (eg, post-treatment, 6-month follow-up, 12-month follow-up).

▶ Method of support (eg, email, face to face, internet, telephone, video conference).

▶ Recruitment setting (eg, clinical, community, mixed).

▶ Type of control condition (eg, no-treatment control, wait-list control, TAU, non-specific factors component control, specific factors component control, active comparator).

▶ Type of dementia (eg, Alzheimer's disease, cerebrovascular/vascular disease, frontotemporal dementia, Lewy body disease or mixed).

▶ Type of psychological intervention (eg, cognitive behavioural therapy, behavioural activation therapy, problem-solving therapy, third-wave cognitive behavioural therapies, interpersonal psychotherapy, psychodynamic therapy, non-directive supportive therapy, life review therapy).

### Sensitivity analyses

Sensitivity analysis regarding the overall effect size for each primary outcome (mental health and psychological well-being for people with dementia or MCI) will be conducted by temporarily removing: (1) each study individually; (2) studies with high attrition (≥30% in at least one trial arm); (3) studies with a small sample size (≤20 in at least one trial arm)[53 54]; and (4) studies rated as high, moderate, and low risk of bias.

### Funnel asymmetry

Funnel plot asymmetry will be examined for possible bias if there is a minimum of 10 studies in any meta-analysis. Estimated effect sizes, taking possible biases into account, will be calculated using the trim and fill procedure,[72] for each primary outcome (mental health and psychological well-being for people with dementia or MCI).

### Intervention acceptability

A deductive framework synthesis will be adopted[33 73] using the Theoretical Framework of Acceptability (TFA)[31] (table 2). TFA constructs will be used as the deductive coding scheme[74] and inductive coding used for relevant data that does not fit with the TFA.[33] Full text included articles will be imported into NVivo V.14 and data from the results sections and relevant online supplemental files

**Table 2** The theoretical framework of acceptability (TFA)*

| TFA construct | Definition |
|---|---|
| Ethically | The extent to which the intervention has a good fit with an individual's value system |
| Affective attitude | Anticipated: How an individual feels about the intervention prior to taking part<br>Experienced: How an individual feels about the intervention after taking part |
| Burden | Anticipated: The perceived amount of effort that is required to participate in the intervention<br>Experienced: The amount of effort that was required to participate in the intervention |
| Opportunity costs | Anticipated: The extent to which benefits, profits or values must be given up to engage in the intervention<br>Experienced: The benefits, profits or values that were given up to engage in the intervention |
| Perceived effectiveness | Anticipated: The extent to which the intervention is perceived to be likely to achieve its purpose<br>Experienced: The extent to which the intervention is perceived to have achieved its intended purpose |
| Self-efficacy | The participant's confidence that they can perform the behaviour(s) required to participate in the intervention |
| Intervention coherence | The extent to which the participant understands the intervention and how it works |

*Adapted from Sekhon et al.[31]

will be coded in accordance with TFA constructs. Initially, a sample of 10% of included texts will be coded by two reviewers independently, with coding discussed in an analysis workshop with a third reviewer to facilitate conceptual consistency and develop a nuanced understanding of the data. Remaining data will be coded by one main reviewer, with analysis workshops held with the second reviewer to discuss and review the coding. Finally, a thematic analysis will be conducted by the main reviewer to identify themes that 'go beyond' the primary data, synthesising findings across studies in relation to review objectives. The thematic analysis will be subsequently discussed in an analysis workshop held with the second reviewer and will be peer-examined by other research team members. Narrative descriptions of each theme will be provided, alongside supporting quotations. A review audit trail will be used to document all changes regarding codes, discussions and decisions to increase trustworthiness.[75]

### Completeness and quality of intervention reporting
To determine the overall completeness and quality of intervention reporting, each item of the TIDieR checklist[36] (table 1) will be scored as: 0 (not reported), 1 (partially reported), or 2 (fully reported), allowing a total summary score ranging from 0 (poor reporting) to 24 (full reporting).[76] Overall completeness of intervention reporting will be determined within each study, as well as across all studies.

### Integration of quantitative and qualitative evidence
In accordance with the convergent segregated approach,[46] quantitative and qualitative findings will be integrated and synthesised once each separate synthesis has been conducted. Integration of quantitative and qualitative findings can result in a greater understanding and aid interpretation of results, compared with only undertaking two separate syntheses without formally linking the two sets of evidence.[46] Findings of each synthesis will be compared and contrasted accordingly[46] and presented either in a configured analysis or narrative analysis if configuration is not possible.[46]

### Patient and public involvement
No patients or public were involved in the design or development of this protocol. However, we have already established a public advisory board consisting of informal caregivers (n=4) with lived experience of caring for a person with dementia. The public advisory board members are aged 44–71 years old and are wives and daughters of people living with dementia with 5–9 years of experience caring for a person with dementia. The public advisory board members will be presented with review findings in an online workshop to help aid interpretation and facilitate the research team to place review findings in context and their perspectives will be incorporated into the results of the review.[77] After presenting review findings, public advisory board members will be asked to discuss: (1) to what extent the framework synthesis of intervention acceptability reflects their own experiences; (2) whether the research team's interpretation of review findings is appropriate and/or whether findings might be interpreted in different ways; and (3) the implications of review findings for people with dementia and MCI and their informal caregivers. The perspectives of public advisory board members' perspectives will be incorporated into the results (ie, reflections on the thematic synthesis) and discussion (ie, interpretation of findings and implications) sections of the results manuscript as relevant.

## DISCUSSION
This is a study protocol for a mixed-methods systematic review that aims to: (1) examine the effectiveness of psychological interventions to improve mental health and psychological well-being in people with dementia or

MCI; (2) examine the effectiveness of these psychological interventions to improve mental health and psychological well-being in informal caregivers; (3) examine potential clinical and methodological moderators associated with effectiveness; (4) explore factors associated with the acceptability of psychological interventions from the perspective of key stakeholders; and (5) examine the completeness and quality of intervention reporting.

This review will extend the existing literature[13–15 18] by including novel aspects such as narrowing the classification and definition of eligible psychological interventions, excluding studies with a high risk of bias, examining clinical and methodological moderators, and exploring intervention acceptability and the completeness and quality of intervention reporting. The planned review has additional strengths. We adopt a mixed-method approach, allowing the synthesis of evidence from multiple sources to address review questions. We follow quality standards informed by the New Cochrane Handbook for Systematic Reviews of Interventions,[42 45] PRISMA 2020 guidelines,[42] and ENTREQ.[41] For example, screening, selection, data extraction, and quality appraisal will be assessed by two researchers independently. While there are several strengths, there are also limitations. Despite the CASP checklist being a recommended tool for assessing the quality of qualitative studies,[78] the checklist does not include a score rating system for scoring the quality of qualitative studies. This may lead to the inclusion of low-quality qualitative studies. However, two reviewers will rate the studies with a third reviewer consulted if a consensus cannot be reached, potentially mitigating this limitation. Due to limited resources, only studies available in English and Swedish will be included, and language bias may be present.[79]

This review will be conducted within phase I (development) of the Medical Research Council framework for developing complex interventions.[80] Results will be used to inform the further adaptation of a psychological intervention targeting depression among people with dementia[81] by identifying important considerations such as barriers and facilitators for intervention use, intervention acceptability, and potential intervention adaptations. Results will also provide important information regarding intervention components and intervention reporting quality and completeness of psychological interventions for people with dementia or MCI. Such findings will be useful for implementation considerations, including healthcare providers delivering psychological interventions for people with dementia and policy-makers.

## ETHICS AND DISSEMINATION

No ethical approval or informed consent will be required as no original data will be collected. Results will be published in an open-access peer-reviewed journal, presented at academic conferences, and disseminated among lay and healthcare professional audiences.

**Author affiliations**
¹Department of Women's and Children's Health, Uppsala University, Uppsala, Sweden
²Clinical Education Development and Research (CEDAR), University of Exeter, Exeter, UK
³Department of Civil and Industrial Engineering, Uppsala University, Uppsala, Sweden
⁴Department of Public Health and Caring Sciences, Uppsala University, Uppsala, Sweden
⁵Department of Medical Sciences, Dalarna University, Falun, Sweden

**Acknowledgements** We would like to thank Agnes Kotka, Librarian at Uppsala University Library for assisting with the development of the electronic search strategy.

**Contributors** FS: methodology, resources, writing–original draft, visualisation, project administration. OB: methodology, writing–review and editing, project administration. PF: conceptualisation, methodology, writing–review and editing, supervision. AB: writing–review and editing, supervision. LvE: writing–review and editing, funding acquisition. ACÅ: writing–review and editing. JW (review guarantor): conceptualisation, methodology, writing–original draft, supervision, project administration, funding acquisition.

**Funding** This work was supported by the Swedish Research Council (Dnr: 2018-02691) and via the Swedish Research Council to U-CARE, a Strategic Research environment (Dnr: 2009–1093).

**Disclaimer** Funders were not involved in the creation, development or publication of this protocol. Funders will not be involved in the conduct, analysis or reporting of the resulting study.

**Competing interests** None declared.

**Patient and public involvement** Patients and/or the public were not involved in the design, or conduct, or reporting, or dissemination plans of this research.

**Patient consent for publication** Not applicable.

**Provenance and peer review** Not commissioned; externally peer reviewed.

**ORCID iDs**
Frida Svedin http://orcid.org/0000-0002-8421-4908
Joanne Woodford http://orcid.org/0000-0001-5062-6798

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
