## [Reviewer comments · BMJ Open]

ARTICLE DETAILS

TITLE (PROVISIONAL)	Effectiveness, acceptability, and completeness and quality of intervention reporting of psychological interventions for people with dementia or mild cognitive impairment: protocol for a mixed-methods systematic review
AUTHORS	Svedin, Frida; Blomberg, Oscar; Farrand, Paul; Brantnell, Anders; von Essen, Louise; Åberg, Anna; Woodford, Joanne

VERSION 1 – REVIEW

REVIEWER	D'Cunha, Nathan University of Canberra Faculty of Health
REVIEW RETURNED	18-Aug-2023

GENERAL COMMENTS	This is a well-written and comprehensive protocol, and an ambitious review. I have only two minor comments: Page 8, Line 272: Suggest stating here that the review design was informed by a research librarian as the person is given credit in the Acknowledgements section which is sufficient. Page 16, Line 483: Please describe in more detail. If the public advisory board are presented with the findings, how will their perspectives be incorporated into the results section of the review? If they are assisting with interpretation, wouldn't this more usefully be incorporated into the discussion of the results?
---

VERSION 1 – AUTHOR RESPONSE

Reviewer 1

Thank you for your very helpful comments. Please find how we have addressed each comment below.

This is a well-written and comprehensive protocol, and an ambitious review. I have only two minor comments:

Page 8, Line 272: Suggest stating here that the review design was informed by a research librarian as the person is given credit in the Acknowledgements section which is sufficient.

We have removed the name of the librarian as follows:

Page 8, line 168-169:

The search strategy was developed alongside a librarian at Uppsala University Library.

Page 16, Line 483: Please describe in more detail. If the public advisory board are presented with the findings, how will their perspectives be incorporated into the results section of the

review? If they are assisting with interpretation, wouldn't this more usefully be incorporated into the discussion of the results?

Thank you for this comment. We have added some additional detail under Patient and Public Involvement concerning how the Public Advisory Board's perspectives will be incorporated into the review:

Page 16, lines 480-490:

The Public Advisory Board members will be presented with review findings **in an online workshop** to help aid interpretation **and facilitate the research team to place review findings in context** and their perspectives will be incorporated into the results of the review.[78] **After presenting review findings, Public Advisory Board members will be asked to discuss (1) to what extent the framework synthesis of intervention acceptability reflects their own experiences; (2) whether the research team's interpretation of review findings is appropriate and/or whether findings might be interpreted in different ways; and (3) the implications of review findings for people with dementia and MCI and their informal caregivers. The perspectives of Public Advisory Board members perspectives will be incorporated into the results (i.e., reflections on the thematic synthesis) and discussion (i.e., interpretation of findings and implications) sections of the results manuscript as relevant.**